Alveolar soft-part sarcoma (ASPS) resembles a mesenchymal stromal progenitor: evidence from meta-analysis of transcriptomic data

Stockwin Luke H. luke.stockwin@gmail.com
Unaffiliated , Frederick , Maryland
Nakai Kenta
Electronic publication date: 2020 Jun 19
Publication date: 2020
Volume: 8
Electronic Location ID: e9394
Received 2020 Apr 16; Accepted 2020 May 29
Copyright: ©2020 Stockwin
Copyright year: 2020
Copyright holder: Stockwin
License: This is an open access article distributed under the terms of the Creative Commons Attribution License, which permits unrestricted use, distribution, reproduction and adaptation in any medium and for any purpose provided that it is properly attributed. For attribution, the original author(s), title, publication source (PeerJ) and either DOI or URL of the article must be cited.
License URL: https://creativecommons.org/licenses/by/4.0/

Keywords: ASPS, Sarcoma, Mesenchymal, Stromal, Microarray, RNA-seq, Meta-analysis, Surfaceome, Genomics, Transcriptome

Funding: The author received no funding for this work.

==============================
Alveolar soft-part sarcoma (ASPS) is an extremely rare malignancy characterized by the unbalanced translocation der(17)t(X;17)(p11;q25). This translocation generates a fusion protein, ASPL-TFE3, that drives pathogenesis through aberrant transcriptional activity. Although considerable progress has been made in identifying ASPS therapeutic vulnerabilities (e.g., MET inhibitors), basic research efforts are hampered by the lack of appropriate in vitro reagents with which to study the disease. In this report, previously unmined microarray data for the ASPS cell line, ASPS-1, was analyzed relative to the NCI sarcoma cell line panel. These data were combined with meta-analysis of pre-existing ASPS patient microarray and RNA-seq data to derive a platform-independent ASPS transcriptome. Results demonstrated that ASPS-1, in the context of the NCI sarcoma cell panel, had some similarities to normal mesenchymal cells and connective tissue sarcomas. The cell line was characterized by high relative expression of transcripts such as CRYAB, MT1G, GCSAML, and SV2B. Notably, ASPS-1 lacked mRNA expression of myogenesis-related factors MYF5, MYF6, MYOD1, MYOG, PAX3, and PAX7. Furthermore, ASPS-1 had a predicted mRNA surfaceome resembling an undifferentiated mesenchymal stromal cell through expression of GPNMB, CD9 (TSPAN29), CD26 (DPP4), CD49C (ITGA3), CD54 (ICAM1), CD63 (TSPAN30), CD68 (SCARD1), CD130 (IL6ST), CD146 (MCAM), CD147 (BSG), CD151 (SFA-1), CD166 (ALCAM), CD222 (IGF2R), CD230 (PRP), CD236 (GPC), CD243 (ABCB1), and CD325 (CDHN). Subsequent re-analysis of ASPS patient data generated a consensus expression profile with considerable overlap between studies. In common with ASPS-1, elevated expression was noted for CTSK, DPP4, GPNMB, INHBE, LOXL4, PSG9, SLC20A1, STS, SULT1C2, SV2B, and UPP1. Transcripts over-expressed only in ASPS patient samples included ABCB5, CYP17A1, HIF1A, MDK, P4HB, PRL, and PSAP. These observations are consistent with that expected for a mesenchymal progenitor cell with adipogenic, osteogenic, or chondrogenic potential. In summary, the consensus data generated in this study highlight the unique and highly conserved nature of the ASPS transcriptome. Although the ability of the ASPL-TFE3 fusion to perturb mRNA expression must be acknowledged, the prevailing ASPS transcriptome resembles that of a mesenchymal stromal progenitor.

Introduction

Alveolar Soft-Part Sarcoma (ASPS) is an extremely rare soft tissue sarcoma of adolescents and young adults (Christopherson, Foote & Stewart, 1952; Paoluzzi & Maki, 2019). ASPS usually manifests as a soft, painless, slow-growing mass and although disease follows an indolent course, it has the potential to metastasize to several sites (Portera Jr et al., 2001). ASPS is characterized by an unbalanced translocation t(X;17)(p11;q25) that fuses the ASPSCR1 and TFE3 genes, generating a fusion protein that drives pathogenesis (Ladanyi et al., 2001). Evidence suggests that the fusion protein accumulates in the nucleus and directs transcriptional activity (Argani et al., 2003; Betschinger et al., 2013; Hirobe et al., 2013). For example, ASPL-TFE3 binds and activates MET transcription, resulting in an overall enhancement in kinase activity in the presence of hepatocyte growth factor (Tsuda et al., 2007). As a consequence, some clinical benefit is being achieved with kinase inhibitors targeting MET (Paoluzzi & Maki, 2019; Schoffski et al., 2018).

Despite this progress, the origin of disease is still the subject of intense speculation (Folpe & Deyrup, 2006). A longstanding hypothesis posits that ASPS has a myogenous origin (Fisher & Reidbord, 1971; Folpe & Deyrup, 2006; Mukai et al., 1983). However, ASPS tumors do not appear to express markers of muscle cell differentiation such as the myogenic nuclear regulatory proteins MyoD1 and myogenin (Gomez et al., 1999; Hoshino et al., 2009; Tallini et al., 1994; Wang et al., 1996). Several transcriptomic studies have also been published that speculate on the origin of disease (Goodwin et al., 2014; Selvarajah et al., 2014; Stockwin et al., 2009; Tanaka et al., 2017). In 2009 we undertook one of the first microarray studies of ASPS and identified expression of several muscle-restricted transcripts (ITGB1BP3/MIBP, MYF5, MYF6, and TRIM63). However, these data were generated using universal RNA as a reference, which may have biased results towards skeletal muscle expressed transcripts (Stockwin et al., 2009). Selvarajah et al. (2014) showed that the transcription factor PAX6 was upregulated in primary ASPS, suggesting a “tentative neural line of differentiation for ASPS”. Goodwin et al. (2014) generated microarray data from a mouse model of ASPS and also human patient samples. These authors speculated that “some mesenchymal progenitor, possibly pericyte/endothelial in character, provides one potential cell of origin”. Similarly, Tanaka et al. (2017) were able to model ASPS through ectopic expression of ASPL-TFE3 in murine embryonic, but not adult, mesenchymal cells. These observations underscore the current lack of clarity with respect to ASPS ontogeny and lend support to the suggestion that ASPS cells represent a “scrambled” phenotype where the ASPL-TFE3 fusion impairs differentiation (Folpe & Deyrup, 2006; Naka et al., 2013).

In 2011, a multi-year study culminated in the development of an ASPS cell line designated ASPS-1 (Kenney et al., 2011). This reagent provided the first opportunity to study ASPS gene expression without interference from contaminating cell types. Microarray data was subsequently generated for ASPS-1 as part of the NCI sarcoma cell line panel (Teicher et al., 2015). In the current study, ASPS-1 data was mined relative to the entire NCI sarcoma cell line panel. These efforts were combined with re-analysis of microarray and RNA-seq studies focusing on ASPS patient samples (Goodwin et al., 2014; Kummar et al., 2013; Stockwin et al., 2009). In this regard, we aimed to unify current publicly available transcriptomic data into a consensus profile that can be used as a basis for exploring disease ontogeny and therapeutic vulnerabilities. Results obtained in this study show that, at the mRNA level, ASPS resembles a mesenchymal stromal cell (MSC).1

Materials and Methods

This study utilized six datasets downloaded from the gene expression omnibus (https://www.ncbi.nlm.nih.gov/geo/). GSE68591 comprises exon expression data (Affymetrix Human Exon 1.0 v2 ST platform) for the NCI sarcoma cell line panel (includes data from ASPS-1, 67 sarcoma lines, and five normal tissues) (Teicher et al., 2015). GSE13433 comprises mRNA expression data for seven ASPS patient tumors analyzed using the Affymetrix U133 plus 2.0 platform (Stockwin et al., 2009). For the analyses of data from GSE13433, additional U133 plus 2.0 control arrays were obtained from GSE17070 (normal skeletal muscle) and GSE118370 (normal lung). GSE32569 comprises a set of U133 plus 2.0 microarrays generated from six patients pre- and post- treatment with Cediranib (Kummar et al., 2013). Lastly, GSE54729 comprises RNA-seq data (HiSeq 2000) from five ASPS patients and three skeletal muscle controls. The overall study design is illustrated in Fig. 1. For experiments involving Affymetrix human Exon 1.0 ST and U133 plus 2.0 arrays differentially expressed genes were identified using the Transcriptome Analysis Console (TAC 4.0, ThermoFisher Scientific) using standard algorithm and comparison settings (RMA normalization, P < 0.05, FDR <0.05, fold change +/-2). The TAC was also used to generate hierarchical clusters using the automated workflow. For RNA-seq data, differential expression values relative to skeletal muscle were determined using the GSE54729_10408R.txt spreadsheet that accompanies the submission. In detail, normalized FKPM values were averaged for the 5 human ASPS samples and the 3 normal human skeletal muscles samples; fold changes were then calculated from these values. In terms of utilities; the GTEX portal (https://www.gtexportal.org/) multi gene query option (https://gtexportal.org/home/multiGeneQueryPage) was used to inform tissue of origin from the top 50 differentially expressed ASPS-1 transcripts. Similarly, the GTEX Top 50 expressed genes search function (https://gtexportal.org/home/topExpressedGenePage) was used to identify genes expressed selectively in skeletal muscle. The Protein Atlas (https://www.proteinatlas.org/) was used to investigate both mRNA and protein expression in normal and cancerous samples for specific transcripts. The in silico surfaceome (http://wlab.ethz.ch/surfaceome/) (Bausch-Fluck et al., 2018) was used to predict the hierarchy of cell surface protein expression for ASPS-1. In detail, a file containing the published human surfaceome (table_S3_surfaceome.xlsx) was downloaded from http://wlab.ethz.ch/surfaceome/ and merged, using MS excel, with the list of differentially expressed ASPS-1 transcripts. Transcripts appearing in both datasets were then extracted and sorted according to ASPS-1 expression. The VENN diagram utility InteractiVenn (http://www.interactivenn.net/) (Heberle et al., 2015) was used to determine the extent of overlap between the 4 different experimental approaches.

Figure 1 Study design.

This study focuses on the analysis of four publicly available GEO gene-expression datasets. GSE68591 comprises exon level expression data for the NCI sarcoma cell line panel, which includes data for the ASPS cell line ASPS-1. The remaining three studies comprise transcriptomic data for ASPS patient samples. GSE13433 comprises Affymetrix U133 plus 2.0 microarray data from our initial gene expression study of seven ASPS patients. GSE32569 uses the same array platform to study ASPS patient sample responses to Cedirinib. Lastly, GSE54729 comprises Illumina HISeq 2000 RNA-seq data for ASPS patient samples generated as part of an ASPS mouse modeling study. These data were re-analyzed using appropriate controls in order to generate a consensus transcriptome and gain insights into ASPS pathobiology.

Results and Discussion

In the first analysis, ASPS-1 exon-level data from the NCI sarcoma cell line panel was analyzed relative to all samples (cancer and normal). Results shown in Table 1 list the top fifty transcripts over-expressed in ASPS-1 relative to the average (Data S1). Results demonstrated that crystallin alpha beta (CRYAB) mRNA showed the highest expression in ASPS-1 relative to the average (227 fold). Metallothionein 1G (MT1G) was the next most elevated (202 fold). Following this were several lower-abundance transcripts coding for C7orf69, Synaptic Vesicle Glycoprotein 2B (SV2B) and Germinal Center-Associated Signaling and Motility-Like Protein (GCSAML). In common with the initial published report of ASPS-1, results also confirmed expression of GPNMB (Kenney et al., 2011). Likewise, ASPS-1 had some of the highest levels of MET, and VEGFR2 in the panel, both of which are previously noted characteristics of disease (Jun et al., 2010; Stockwin et al., 2009; Tsuda et al., 2007). This analysis served to confirm the ASPS origin of ASPS-1 as outlined in Kenney et al. (2011).

Following this, ASPS-1 was subjected to hierarchical clustering relative to the entire NCI sarcoma cell line panel (Fig. 2A). Results showed that ASPS-1 was an outlier that did not closely align with any lines/tissues in the panel. Those specimens with the nearest similarity are shown in Fig. 2B; this includes normal cell/tissues such as knee articular chondrocytes, dermal fibroblasts, skeletal muscle, mesenchymal stem cells and osteoblasts. Cancer line relatives included the spindle cell sarcoma Hs 132.T, the fibrosarcomas Hs 93.T and Hs 414.T along with the chondrosarcoma Hs 819. In order to define which changes were guiding this clustering, several breakout analyses were performed. Over-expressed transcripts found in the nearest cluster group and ASPS-1 included GPNMB, CRYAB, FABP3, and CTSK, which are markers of mesenchymal cells (Debnath et al., 2018; Kulterer et al., 2007; Wang Jr et al, 2014; Yu et al., 2016). Transcripts expressed only in ASPS-1 relative to the nearest cluster included SEPT3, C7orf69, MT1G, and ACP5, many of which are implicated in the differentiation of mesenchymal stromal cells (Dohi et al., 2005; Hayman et al., 2000; Moller et al., 2018; Tan et al., 2015). Lastly, the nearest cluster group could be distinguished from ASPS-1 through higher expression of the canonical fibroblast markers GREM1, LOX, THY1, and POSTN (Hortells, Johansen & Yutzey, 2019; Jiang & Rinkevich, 2018; Karagiannis et al., 2012). These data point towards a mesenchymal stromal origin that had not undergone significant fibroblast lineage specialization.

Table 1 The top fifty over-expressed genes in ASPS-1 relative to the average of the NCI sarcoma cell line panel.

Gene symbol	Affy ID	Description	FC ASPS-1 vs. Av	
CRYAB; FDXACB1	3391149	crystallin alpha B; ferredoxin-fold anticodon binding domain containing 1	227.1	
MT1G	3692999	metallothionein 1G	202.9	
C7orf69	3000905	chromosome 7 open reading frame 69	202.0	
GCSAML	2390102	germinal center-associated, signaling and motility-like	177.6	
SV2B	3608638	synaptic vesicle glycoprotein 2B	132.6	
ADGRL4	2419432	adhesion G protein-coupled receptor L4	129.9	
PLA2G7	2955827	phospholipase A2, group VII	93.5	
SULT1C2	2498911	sulfotransferase family 1C member 2	91.3	
SLN	3389954	sarcolipin	88.9	
CFAP61	3878972	cilia and flagella associated protein 61	85.2	
PPEF1	3970714	protein phosphatase, EF-hand calcium binding domain 1	82.4	
ACP5	3851072	acid phosphatase 5, tartrate resistant	79.8	
CD36	3010503	CD36 molecule (thrombospondin receptor)	79.1	
PPARGC1A	2763550	peroxisome proliferator-activated receptor gamma, coactivator 1 alpha	78.9	
ASB11	4000485	ankyrin repeat and SOCS box containing 11, E3 ubiquitin protein ligase	78.1	
BMP5	2958172	bone morphogenetic protein 5	74.9	
PRUNE2	3210616	prune homolog 2 (Drosophila)	72.3	
SUCNR1	2648098	succinate receptor 1	66.5	
PSG9	3864286	pregnancy specific beta-1-glycoprotein 9	66.0	
CKMT2	2818035	creatine kinase, mitochondrial 2 (sarcomeric)	61.7	
DPP4	2584018	dipeptidyl-peptidase 4	61.2	
ABCB1	3060182	ATP binding cassette subfamily B member 1	59.7	
SCIN	2990404	scinderin	58.7	
FABP3	2404418	fatty acid binding protein 3, muscle and heart	55.8	
PRUNE2	3210497	prune homolog 2 (Drosophila)	55.7	
SLC27A2	3593575	solute carrier family 27 (fatty acid transporter), member 2	54.7	
SLCO4C1	2869096	solute carrier organic anion transporter family, member 4C1	51.9	
PSG11,5,4,2	3863929	pregnancy specific beta-1-glycoprotein 11,5,4,2	50.4	
PRLR	2853102	prolactin receptor	47.9	
NPY6R	2830450	neuropeptide Y receptor Y6 (pseudogene)	44.6	
ANXA3	2732844	annexin A3	43.2	
TRPC7	2876793	transient receptor potential cation channel, subfamily C, member 7	42.9	
CD5L	2439138	CD5 molecule-like	41.0	
AKR1C2	3274758	aldo-keto reductase family 1, member C2	40.9	
GPNMB	2992814	glycoprotein (transmembrane) nmb	40.3	
IL13RA2	4018729	interleukin 13 receptor, alpha 2	39.1	
LRRC39	2425173	leucine rich repeat containing 39	38.8	
CST1	3901361	cystatin SN	37.4	
CDH7	3792273	cadherin 7, type 2	36.8	
DOK5	3889833	docking protein 5	35.1	
SEPT3; WBP2NL	3947227	septin 3; WBP2 N-terminal like	34.8	
GCNT3	3596147	glucosaminyl (N-acetyl) transferase 3, mucin type	34.8	
ENPP5	2955673	ectonucleotide pyrophosphatase/phosphodiesterase 5 (putative)	34.7	
DOK3	2888879	docking protein 3	34.2	
LCP1	3512874	lymphocyte cytosolic protein 1 (L-plastin)	34.0	
CDA	2324084	cytidine deaminase	33.4	
KLHL4	3983324	kelch-like family member 4	33.3	
CTSK	2434609	cathepsin K	31.5	
LIPC	3595691	lipase, hepatic	31.0	
RPSA	3827218	ribosomal protein SA	30.0	

Figure 2 Analysis of ASPS-1 transcriptomic data.

(A) Heirarchical clustering of the NCI sarcoma cell line panel exon array data (B) tissue or tumor derivation for cluster nearest ASPS-1. (C) GTEX tissue of origin analysis for top fifty ASPS-1 transcripts.

The Genotype-Tissue Expression (GTEx) portal (https://www.gtexportal.org/) was then used in order to determine whether the ASPS-1 expression data suggested a tissue of origin. Results obtained using the top fifty over-expressed transcripts suggested that cardiac/skeletal muscle was a likely origin through expression of transcripts such as CRYAB, FABP3, SLN, and CKMT2 (Fig. 2C). The GTEX database also lists the top 50 transcripts expressed by normal skeletal muscle, where 28 (e.g., creatine kinase M-Type CKM and myoglobin MB) have a considerable degree of muscle-specificity. Results showed that only two transcripts from this set, EEF1A2 and SLN are over-expressed in ASPS-1. As a consequence, we then undertook an expansive study of whether ASPS-1 had any hallmarks of myogenic differentiation. Figure 3 plots raw expression data for all cell lines in the sarcoma panel, where lines are grouped according to histology. Results showed that ASPS-1, unlike some rhabdosarcoma lines, did not express myogenic regulatory factor mRNAs including MYF5, MYF6, MYOD1, MYOG, and similarly did not express PAX3 and PAX7. Taken together these observations suggest that although ASPS-1 has hallmarks of a muscle resident cell, it had not undergone myogenic differentiation.

Figure 3 Expression of myogenesis-related transcripts in ASPS-1 relative to the other sarcoma cell lines.

Cell lines in the NCI sarcoma panel were segregated according to disease type and the average of exon expression data plotted for transcripts encoding myogenesis-related transcription factors and muscle structural proteins. ASPS-1 data is shown first; then RMS, Rhabdomyosarcoma; RT, Rhabdoid tumor; LMS, Leiomyosarcoma; CS, Chondrosarcoma; FS, Fibrosarcoma; EWS, Ewings Sarcoma; OS, Osteosarcoma; SS, Synovial Sarcoma; US, Uterine Sarcoma; SNOS, Sarcoma not otherwise specified; GCS, Giant cell sarcoma; MNS, Malignant peripheral nerve sheath; LPS, Liposarcoma; SPS, Spindle cell sarcoma; EP, Epithelioid; NC, Normal cells; SkMc, skeletal muscle cells. For transcripts; MYF5, Myogenic Factor 5, MYF6, Myogenic Factor 6, MYOD1, Myogenic Differentiation 1, MYOG = Myogenin, PAX3, Paired Box 3, PAX7 = Paired Box 7. Structural proteins; DES, desmin, NEB, nebulin, TNNT1, Troponin T1 - Slow Skeletal Type, TRIM63, Tripartite Motif Containing 63, TTN, titin and MSTN, Myostatin. Transcripts evaluated but not shown included; EYA1, LBX1, MEF2B, MEOX2, MITF, MSX1, PITX1, SIM2, SIX1, SIX4, TFE3 and TFEB.

These data also provide a unique opportunity to assess the potential cell surface phenotype of ASPS-1. Bausch-Fluck et al. (2018) identified 2886 proteins that are known, or are predicted by machine learning, to be expressed on the cell surface . Here, ASPS-1 raw microarray data was filtered for these targets and the list sorted in terms of expression. The resultant ASPS-1 ‘surfaceome’ is shown in Data S2. As could be anticipated, GPNMB was the highest expressed mRNA; followed by novel surface makers such as the glutamate transporter SLC38A1 and the amyloid beta (A4) precursor protein APP. In terms of CD antigens, the following mRNAs were highly expressed in ASPS-1; CD9 (TSPAN29), CD26 (DPP4), CD49C (ITGA3), CD54 (ICAM1), CD63 (TSPAN30), CD68 (SCARD1), CD130 (IL6ST), CD146 (MCAM), CD147 (BSG), CD151 (SFA-1), CD166 (ALCAM), CD222 (IGF2R), CD230 (PRP), CD236 (GPC), CD243 (ABCB1), and CD325 (CDHN).

Many of these observations are also compatible with a mesenchymal stromal cell. For example, CD9 (TSPAN29) and CD243 (ABCB1), although widely expressed, are found to varying degrees on MSC (Islam et al., 2005; Kim et al., 2007). CD49C (ITGA3) and CD151 (SFA-1) are both markers of chondrogenic differentiation in MSC (Grogan et al., 2007; Lee et al., 2009a) . Expression of CD54 (ICAM1) can be induced in MSC (Ren et al., 2010), CD63 (TSPAN30) is expressed by bone marrow MSC (McBride et al., 2017), CD68 (SCARD1) expression has been shown on MSC from human umbilical cord (Rocca, Anzalone & Farina, 2009) and CDH2 is a regulator of mesenchymal stem cell fate (Alimperti & Andreadis, 2015). Exosomes expressing Basigin, BSG (CD147), have been shown to promote angiogenesis in MSC (Vrijsen et al., 2016). CD147 is also a major constituent of the pre-crystalline granules present in ASPS (Ladanyi et al., 2002). Expression of Melanoma Cell Adhesion Molecule, MCAM (CD146), mRNA provides strong evidence for an MSC derivation, with several studies demonstrating an important role for this molecule in MSC maintenance and differentiation (Covas et al., 2008; Espagnolle et al., 2014; Jin et al., 2016; Stopp et al., 2013). Likewise, Activated Leukocyte Cell Adhesion Molecule (ALCAM, CD166), is a recognized marker of MSC and implicated in osteogenesis (Bruder et al., 1998; Hu et al., 2016).

In summary, the results presented here demonstrate that the ASPS-1 transcriptome is unique amongst the NCI sarcoma panel, where the closest relatives are normal mesenchymal cells and connective tissue sarcomas. Although ASPS-1 has an expression signature with some similarity to skeletal/cardiac muscle tissue, markers of myogenesis were not detected in this cell line. Furthermore, the ASPS-1 surfaceome does not immediately speak to a tissue derivation but suggests an undifferentiated mesenchymal state.

The next phase of the project involved re-analyzing microarray and RNA-seq data from ASPS tumor resections. GSE13433 comprises microarray data (Affymetrix U133 plus 2.0) for seven patients with primary or metastatic ASPS (Stockwin et al., 2009). In the original study, universal RNA (representing a collection of adult human tissues) was used as a reference. However, patient samples 1,3, 5 and 6 were obtained from skeletal muscle biopsies whereas samples 4 and 7 were isolated from lung. As a consequence, microarray data from normal skeletal muscle and lung represent more appropriate controls. Therefore, skeletal muscle arrays were obtained from GSE17070 and normal lung samples from GSE118370. Patient 2 data, derived from the mandible, was excluded from the analysis for lack of an appropriate control. Two lists of differentially expressed transcripts were then generated for patients 1,3,5,6 vs. skeletal muscle and 4,7 vs. normal lung. Results from these two experiments were largely concordant with the profiles obtained in the original study (Stockwin et al., 2009). However, in Stockwin et al. muscle-differentiation associated transcripts ITGB1BP3/MIBP, MYF5 and MYF6 were identified as overexpressed. In our analysis, only MYF6 was identified, and only in the experiment involving patients 4 and 7 vs. normal lung; supporting our inference that the published study over-emphasized myogenic differentiation in patient ASPS.

GSE32569 is a similar ASPS dataset where U133 plus 2.0 microarrays were generated from patients treated with Cediranib (Kummar et al., 2013). We undertook to use this data to generate a list of differentially expressed transcripts from pre-treatment arrays relative to GSE17070 skeletal muscle samples. Results again showed a similar profile to that obtained from the analysis of GSE13433. The final experiment was performed using RNA-seq data from GSE54729. In this published study, data was generated from five human ASPS tumor samples in order to compare the transcriptome with five mouse tumors generated through ectopic expression of ASPSCR1-TFE3 (Goodwin et al., 2014). Here, FKPM values for the five human tumors and three skeletal muscle controls were used to generate a list of differentially expressed transcripts. The top fifty upregulated transcripts generated from each of the four experiments using GSE13433, GSE32569 and GSE54729 are shown in Table 2. Meta-analysis of data from these four in vivo studies had considerable overlap, emphasizing the consistent upregulation of mRNAs such as GPNMB, ABCB5, PSG9, CYP17A1, PRL, SULT1C2, and SV2B. As with ASPS-1, over-expression of myogenic regulatory factor mRNA was not consistently seen in any of the experiments involving patient samples (results not shown). Lastly, lists of differentially expressed genes from both ASPS-1 and patient sample experiments were combined (at a five-fold cut-off 2 ), and a VENN diagram generated in order to determine the extent of overlap (Fig. 4). Results demonstrated that twenty-five transcripts were elevated in all of the meta-analyses, whereas seventy-three were expressed at the intersection between all in vivo analyses.

Table 2 The top fifty upregulated transcripts generated from each of the four experiments utilizing ASPS patient data from GSE13433, GSE32569 and GSE54729.

GSE13433 Patients 1,3,5,6 vs. Skeletal muscle	GSE13433 Patient 4,7 vs. Normal lung	GSE32569 Pre-treatment ASPS vs. Skeletal muscle	GSE54729 Patients vs. Skeletal muscle	
AFFY ID	FC	GENE ID	AFFY ID	FC	GENE ID	AFFY ID	FC	GENE ID	Ensembl ID	FC	Gene ID	
205502_at	639.7	CYP17A1	207733_x_at	608.6	PSG9	211470_s_at	573.8	SULT1C2	ENSG00000159871	100.9	LYPD5	
212992_at	531.8	AHNAK2	209594_x_at	601.3	PSG9	205502_at	420.6	CYP17A1	ENSG00000183979	423.1	NPB	
205445_at	434.1	PRL	205445_at	572.3	PRL	1554018_at	375.0	GPNMB	ENSG00000198203	237.6	SULT1C2	
240717_at	419.0	ABCB5	205502_at	537.1	CYP17A1	205342_s_at	367.4	SULT1C2	ENSG00000148795	1700.4	CYP17A1	
205342_s_at	338.7	SULT1C2	1555786_s_at	510.2	LINC00520	210809_s_at	365.1	POSTN	ENSG00000146678	200.6	IGFBP1	
201850_at	258.8	CAPG	206224_at	467.6	CST1	238720_at	351.1	LOC101927057	ENSG00000172179	357.1	PRL	
211470_s_at	252.9	SULT1C2	223572_at	437.6	HHATL	210587_at	341.2	INHBE	ENSG00000169006	142.2	NTSR2	
238720_at	248.1	LOC101927057	240717_at	346.4	ABCB5	206214_at	262.4	PLA2G7	ENSG00000101197	113.9	BIRC7	
1554018_at	240.8	GPNMB	208555_x_at	322.5	CST2	212992_at	259.6	AHNAK2	ENSG00000170369	345.4	CST2	
205302_at	207.6	IGFBP1	236972_at	277.8	TRIM63	1565162_s_a	241.7	MGST1	ENSG00000100167	143.8	SEPT3	
204638_at	198.2	ACP5	1553663_a_at	259.2	NPB	229831_at	228.7	CNTN3	ENSG00000204632	396.4	HLA-G	
206899_at	170.8	NTSR2	205302_at	255.8	IGFBP1	200832_s_at	225.4	SCD	ENSG00000146070	147.4	PLA2G7	
210587_at	170.4	INHBE	221051_s_at	200.5	NMRK2	206899_at	168.5	NTSR2	ENSG00000110492	313.0	MDK	
212805_at	157.6	PRUNE2	206239_s_at	192.8	SPINK1	205302_at	159.6	IGFBP1	ENSG00000227925	134.1	LOC101929771	
209875_s_at	150.3	SPP1	210587_at	184.3	INHBE	227180_at	157.3	ELOVL7	ENSG00000170373	455.2	CST1	
1557636_a_at	145.1	C7orf57	206899_at	171.9	NTSR2	219648_at	149.3	MREG	ENSG00000225328	118.5	LINC01594	
210809_s_at	144.4	POSTN	205551_at	169.9	SV2B	210397_at	148.0	DEFB1	ENSG00000118785	260.1	SPP1	
221577_x_at	134.0	GDF15	219106_s_at	149.9	KLHL41	209875_s_at	147.2	SPP1	ENSG00000139269	137.6	INHBE	
221008_s_at	133.2	ETNPPL	206799_at	146.3	SCGB1D2	557636_a_a	145.3	C7orf57	ENSG00000102575	190.9	ACP5	
206214_at	130.1	PLA2G7	229052_at	143.1	ANKRD23	205825_at	139.0	PCSK1	ENSG00000185518	114.7	SV2B	
209035_at	122.3	MDK	221008_s_at	142.4	ETNPPL	219073_s_at	138.2	OSBPL10	ENSG00000205336	129.2	ADGRG1	
202450_s_at	121.7	CTSK	205342_s_at	125.6	SULT1C2	205343_at	132.8	SULT1C2	ENSG00000185567	246.4	AHNAK2	
200832_s_at	121.0	SCD	212805_at	124.7	PRUNE2	555778_a_a	127.5	POSTN	ENSG00000136235	2756.8	GPNMB	
212806_at	119.7	PRUNE2	1564758_at	115.7	LOC643659	221008_s_at	126.1	ETNPPL	ENSG00000042493	172.1	CAPG	
230067_at	116.8	FAM124A	229831_at	114.5	CNTN3	231736_x_at	116.1	MGST1	ENSG00000143387	1718.2	CTSK	
208555_x_at	116.8	CST2	209738_x_at	113.4	PSG6	212805_at	115.1	PRUNE2	ENSG00000030582	707.5	GRN	
225275_at	115.8	EDIL3	233389_at	107.5	CFAP61	558378_a_a	108.4	AHNAK2	ENSG00000107317	354.1	PTGDS	
229831_at	112.1	CNTN3	212992_at	106.9	AHNAK2	218292_s_at	98.4	PRKAG2	ENSG00000106617	161.8	PRKAG2	
227404_s_at	111.2	EGR1	233238_s_at	106.0	CTB-12O2.1	221577_x_at	94.6	GDF15	ENSG00000216490	275.5	IFI30	
212841_s_at	109.4	PPFIBP2	1569072_s_at	102.9	ABCB5	218404_at	94.0	SNX10	ENSG00000183696	183.0	UPP1	
216834_at	108.8	RGS1	206994_at	87.9	CST4	233748_x_at	90.8	PRKAG2	ENSG00000110092	192.1	CCND1	
208792_s_at	106.1	CLU	239205_s_at	84.7	CR1; CR1L	224918_x_at	89.1	MGST1	ENSG00000212443	410.2	SNORA53	
223362_s_at	105.2	SEPT3.	217871_s_at	84.7	MIF	212070_at	86.0	ADGRG1	ENSG00000185585	105.1	OLFML2A	
208791_at	92.7	CLU	226086_at	83.8	SYT13	205551_at	85.1	SV2B	ENSG00000130203	285.1	APOE	
1558846_at	92.7	PNLIPRP3	213175_s_at	82.5	SNRPB	244444_at	84.5	PKD1L2	ENSG00000111412	119.3	C12orf49	
230746_s_at	92.3	N/A	221523_s_at	81.9	RRAGD	208965_s_at	83.6	IFI16	ENSG00000206503	1571.8	HLA-A	
218292_s_at	89.1	PRKAG2	243167_at	77.0	ABCB5	208146_s_at	82.8	CPVL	ENSG00000106066	239.3	CPVL	
1565162_s_at	88.4	MGST1	206372_at	74.4	MYF6	226847_at	82.8	FST	ENSG00000138131	108.9	LOXL4	
205825_at	83.7	PCSK1	209875_s_at	70.5	SPP1	223484_at	80.3	C15orf48	ENSG00000118508	104.3	RAB32	
226372_at	82.7	CHST11	244444_at	67.3	PKD1L2	234983_at	78.9	C12orf49	ENSG00000174080	454.1	CTSF	
202503_s_at	82.6	KIAA0101	205862_at	65.9	GREB1	240717_at	78.3	ABCB5	ENSG00000169116	207.1	PARM1	
205343_at	82.0	SULT1C2	222379_at	65.8	KCNE4	229177_at	78.1	C16orf89	ENSG00000120885	187.0	MIR6843	
205551_at	81.7	SV2B	1554371_at	60.2	PKD1L2	205844_at	75.8	VNN1	ENSG00000214435	114.9	AS3MT	
1569072_s_at	81.5	ABCB5	205825_at	58.2	PCSK1	238376_at	75.5	LOC100505564	ENSG00000130208	134.3	APOC1	
227180_at	79.8	ELOVL7	222714_s_at	55.6	LACTB2	205445_at	73.0	PRL	ENSG00000100644	335.4	HIF1A	
231736_x_at	79.5	MGST1	218619_s_at	54.6	SUV39H1	242340_at	71.7	N/A	ENSG00000135047	458.4	CTSL	
202037_s_at	76.5	SFRP1	236523_at	54.3	LOC285556	204285_s_at	71.3	PMAIP1	ENSG00000144136	134.6	SLC20A1	
219648_at	74.5	MREG	1557636_a_at	53.7	C7orf57	204466_s_at	71.3	SNCA	ENSG00000101846	109.4	STS	
206685_at	71.8	HCG4	212070_at	52.9	ADGRG1	203767_s_at	70.7	STS	ENSG00000111775	242.3	COX6A1	
210397_at	71.0	DEFB1	204830_x_at	52.8	PSG5	222872_x_at	70.5	NABP1	ENSG00000089101	164.4	CFAP61	

An exploration of the twenty-five conserved transcripts in the context of stem cell biology provides further insights into MSC lineage potential. For example; angiopoietin Like 2 (ANGPTL2) is a regulator of stem cell adipogenesis, chondrogenesis and osteogenesis (Takano et al., 2017; Tanoue et al., 2018). Expression of Cathepsin K (CTSK) is compelling given that in mice CTSK-mGFP cells label the periosteal mesenchyme and have been used to identify periosteal stem cells (Debnath et al., 2018). As previously noted, Dipeptidyl Peptidase 4 (DPP4), also known as CD26, marks mesenchymal preadipocyte progenitors (Merrick et al., 2019). Glycoprotein Nmb (GPNMB) should be recognized as the prototypic cell surface marker for ASPS. As stated, GPNMB is recognized as a marker of mesenchymal cells (Kuci et al., 2019). Interestingly, within protein atlas, the cell line designated ‘ASC diff’, a differentiated adipose-derived mesenchymal stem cell line has the highest expression of GPNMB and also expresses TRIM63, CRYAB, FABP3, and CTSK. These observations would appear to favor the concept that ASPS resembles an MSC capable of adipogenic, chondrogenic or osteogenic differentiation.

Several inferences can also be made for the seventy-three conserved transcripts identified in all ASPS patient experiments. The multi-drug resistance transporter ABCB5, in addition to being expressed by melanoma, also defines a subset of MSC in the cornea and skin (Frank et al., 2003; Ksander et al., 2014; Vander Beken et al., 2019). The hormone prolactin (PRL) has been shown to stimulate proliferation of MSC and also to direct chondrogenic and ostegenic differentiation (Ogueta et al., 2002; Seriwatanachai, Krishnamra & Charoenphandhu, 2012; Surarit, Krishnamra & Seriwatanachai, 2016). Increased expression of the growth factor midkine (MDK) has been noted in previous ASPS gene expression studies and is an MSC survival factor (Stockwin et al., 2009; Zhao et al., 2014). Upregulation of hypoxia-related transcripts such as HIF1A suggests that this pathway is active in ASPS and, although ubiquitous, HIF1A plays an important role in the control of multipotency for MSC (Palomaki et al., 2013). It was similarly interesting that the ASPS patient experiments showed increased expression of THY1 (CD90) relative to control samples. This target is regarded as a classical marker of MSC and has recently been shown to promote osteogenic differentiation over an adipogenic fate (Saalbach & Anderegg, 2019). In summary, re-analysis of microarray and RNA-seq data for ASPS patient samples yielded transcriptomes with considerable overlap between studies irrespective of platform technology; and the final consensus ASPS transcriptome resembles an undifferentiated mesenchymal stromal cell.

Figure 4 VENN diagram showing overlap between analyses.

Lists of over-expressed transcripts (five-fold cut off) were used to determine extent of overlap between the five datasets. The number of differentially expressed transcripts at five-fold is underlined. Callouts show the 25 transcripts over-expressed in all experiments and the 73 found in all ASPS patient analyses.

Conclusions

Alveolar-soft part sarcoma is an example of a malignancy that has, despite several immunohistochemical and genomics studies, evaded classification (Fisher & Reidbord, 1971; Folpe & Deyrup, 2006; Gomez et al., 1999; Goodwin et al., 2014; Hoshino et al., 2009; Mukai et al., 1983; Selvarajah et al., 2014; Stockwin et al., 2009; Tallini et al., 1994; Tanaka et al., 2017; Wang et al., 1996). This study was prompted by the public release of exon expression data for the cell line ASPS-1, which offers a unique opportunity to study ASPS in isolation (Kenney et al., 2011; Teicher et al., 2015). We were similarly interested in revisiting the genomic studies of ourselves and others to generate a consensus expression profile independent of platform technology.

The central finding of the current study was that the ASPS transcriptome is indicative of an undifferentiated mesenchymal stromal cell (MSC). Specifically, The ASPS-1 cell line exhibited a mesenchymal expression signature, where expression data clustered with normal and malignant mesenchymal cells within the NCI sarcoma cell line panel. The ASPS-1 surfaceome was similarly suggestive of an undifferentiated mesenchymal cell. Generation of an ASPS consensus transcriptome from previously reported patient studies highlighted the importance of targets such as GPNMB, ABCB5, CSTK, DPP4, BSG, ALCAM, PRL, and CDHN; all of which were consistent with an undifferentiated MSC. Conversely, the ASPS transcriptome lacked expression of myogenesis-related genes and did not feature transcripts indicative of neural, pericyte or endothelial differentiation.

MSC are found in most tissues, these cells are capable of multipotent differentiation into bone, muscle, cartilage, adipocytes, marrow stromal cells, tenocytes, fibroblasts, endothelial and neural cells (Caplan, 2007; Pittenger et al., 2019). Tissues maintain a pool of MSC, with varying degrees of specialization, ready to dynamically replenish differentiated cells in response to signals associated with growth, homeostasis or damage (Rubenstein et al., 2020). Prior to this study, ASPL-TFE3 had already been shown to immortalize embryonic mesenchymal cells (Tanaka et al., 2017). The suggestion that ASPS resembles a mesenchymal stromal progenitor provides a plausible explanation for the failure of previous studies to pinpoint cellular origin, given that the cell retains an undifferentiated state. Evidence from this study favors an MSC capable of adipogenic, osteogenic or chondrogenic differentiation, but not necessarily at the exclusion of other lineages.

If an MSC origin for ASPS could ultimately be confirmed, there would be important consequences for therapeutic development. Foremost is the suggestion that ASPS growth may be inhibited by factors that promote MSC differentiation. For example, several high-throughput studies have identified clinically relevant small molecules capable of promoting or inhibiting differentiation of MSC (Brey et al., 2011; Huang et al., 2008). Re-screening these compounds for their ability to inhibit the growth of ASPS-1 may yield clinically tractable candidates for ASPS treatment. From the authors perspective, the effect of HDAC inhibitors, steroids and retinoids on ASPS-1 growth are of particular interest (Lee et al., 2009b; Salloum, Rubin & Marra, 2013).

The findings of this study have a central caveat; all speculation regarding cellular origin must be moderated until the inference of ASPL-TFE3 is removed. Given the ability of this fusion protein to re-direct transcription, the observed transcriptomes may mask the true cellular origin. For example, GPNMB, CRYAB, CYP17A1, SULT1C2, UPP1 and SV2B have been shown to be up-regulated following expression of ASPL-TFE3 in inducible 293 cells (Kobos et al., 2013). Therefore, the current study only suggests that ASPS resembles an MSC and no firm conclusion can be made regarding origin. A straightforward approach to address this central question involves generating an ASPL-TFE3 knockout in ASPS-1 perhaps with re-introduction of wild-type TFE3 to maintain viability. The resultant line could then be characterized by RNA-seq and FACS phenotyping. These experiments could be accompanied by the addition of defined media to determine whether differentiation can be directed toward specific MSC lineages. In the interim, the data presented here provides a unified picture of ASPS mRNA expression, where considerable similarity with mesenchymal stromal progenitors is evident.

Supplemental Information

Supplemental Information 1 PRISMA checklist

Click here for additional data file.

Supplemental Information 2 Raw data for ASPS-1 expression relative to the average of all samples in the NCI sarcoma cell line dataset (2 fold cut-off)

Click here for additional data file.

Supplemental Information 3 Raw data from surfaceome analysis of ASPS-1

Click here for additional data file.

Supplemental Information 4 Raw data from all experiments conducted in this report, extending to concatenation of all data into a single spreadsheet to allow any one gene to be investigated across all analyses

Click here for additional data file.

I would like to thank Dr. Francesco Tomassoni, Dr. C. Andrew Stewart and Marie Stockwin, PAC for scientific and editorial input.

Additional Information and Declarations

Competing Interests

Author Contributions

Human Ethics

Data Availability

1 Throughout this report the abbreviation MSC is used interchangeably for “mesenchymal stromal cells” and “mesenchymal stem cell”, although the former is preferred.

2 Data S3 comprises low-stringency (2-fold) expression data so that any gene of interest can be analyzed for expression over the entire set of experiments.

Luke H. Stockwin is a current employee of Leidos Biomedical Research Inc., however, all of the research conducted in this article was conducted off-site, on an extra-curricular basis, using non-company equipment and was self-financed.

Luke H. Stockwin conceived and designed the experiments, performed the experiments, analyzed the data, prepared figures and/or tables, authored or reviewed drafts of the paper, and approved the final draft.

The following information was supplied relating to ethical approvals (i.e., approving body and any reference numbers):

All IRB approvals are contained within the respective publications cited as a basis for this study.

The following information was supplied regarding data availability:

The raw data are available in Data S1–S3.

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
