# Peer review of "Alveolar soft-part sarcoma (ASPS) resembles a mesenchymal stromal progenitor: evidence from meta-analysis of transcriptomic data"

_PeerJ, doi:10.7717/peerj.9394_

## Round 0.1 · original submission · Minor Revisions

Your manuscript has been reviewed by two experts in the field. As you can see from their comments below, both of them appreciate the value of this work but raise several minor points for further improvement. Particularly, it seems important to add more details in the Materials and Methods section. Please read the comments carefully and revise the manuscript accordingly.

·

Basic reporting

no comment

Experimental design

no comment

Validity of the findings

no comment

Additional comments

The author reports here that ASPS resembles a mesenchymal stromal progenitor at the mRNA level. The study is well illustrated and documented, and the conclusions are certainly supported by the data presented. However, some parts of the manuscript need revision, before it can be accepted for publication.

#1 Materials and methods are too short and not sufficient to understand.
In Table 1, the author has listed the top 50 transcripts overexpressed in the ASPS-1 cell line relative to the average. Is it an appropriate analysis method to use the average value of completely different samples? A bias exists in the number of cell lines. The number of Ewing sarcoma cell lines is 22; on the other hand, the number of DDLS cell lines is only 2. The average value reflects the majority of data (Ewing sarcoma gene expression value). The author has to better compare with ASPS-1 individually for each cell line type.

#2 In Figure 1A, cell lines from the same tumor are grouped into separate clusters. For example, fibrosarcoma and osteosarcoma cell lines are classified into various clusters. The author should check the bootstrap value to confirm the reliability of the branches of the phylogenetic tree. The author may also attempt nonhierarchical clustering, such as PCA.

#3 In Figure 3, the letters are too small and the graph is complicated. The unit of the vertical axis of the graph is not listed.

Reviewer 2 ·

Basic reporting

This study investigated the many genes expression levels in ASPS compared to other sarcoma cells, normal tissues etc. They showed ASPS transcriptome resembles that of mesenchymal stromal progenitor. Therefore, their study based on the data from gene expressions, I thought that it had a certain level of reliability.

Experimental design

Experimental design had no problem. I thought they should do the experiments such as RNAi for comparison between the gene expressions of ASPL-TFE3 knockdown and not. But they commented about the point in the Conclusions. ASPS sarcoma cells have chromosomal translocation of ASPL-TFE3, so, in the cells, various changes could be occurred. So, it is very interesting that they investigate the changes of gene expressions based of RNAi experiments (ASPL-TFE3 knockdown).

Validity of the findings

Their findings had a certain level of reliability.

There are several points that need minor changes as followings;
・In line 53, ASPSCR should be changed to ASPSCR1.
・In line 309, ASPL-TFE should be change to ASPL-TFE3.
・Author should use the unified abbreviation such as MSC in line293-302.
・In Figure2, Table1, 2, author should correct the spelling of “fifty”.

Additional comments

I thought this study is acceptable for this journal. ASPS sarcoma cells have chromosomal translocation of ASPL-TFE3, so, in the cells, various changes could be occurred. So, it is very interesting that they investigate the changes of gene expressions based of RNAi experiments (ASPL-TFE3 knockdown).

---

## Round 0.2 · accepted · Accept

One of the original reviewers as well as myself confirm that the revision has been appropriately done.

Reviewer 2 ·

Basic reporting

I have no comment.

Experimental design

I have no comment.

Validity of the findings

I have no comment.

Additional comments

Authors change the points of reviewers, and this article is acceptable for this journal.